# Feasibility and acceptability of early infant screening for sickle cell disease in Lagos, Nigeria—A pilot study

Esther O. Oluwole[1]*, Titilope A. Adeyemo[2], Gbemisola E. Osanyin[3], Oluwakemi O. Odukoya[1], Phyllis J. Kanki[4], Bosede B. Afolabi[3]

1 Department of Community Health and Primary Care, College of Medicine, University of Lagos, Lagos, Nigeria, 2 Department of Haematology and Blood Transfusion, College of Medicine, University of Lagos, Lagos, Nigeria, 3 Department of Obstetrics and Gynaecology, College of Medicine, University of Lagos, Lagos, Nigeria, 4 Department of Immunology and Infectious Disease, Harvard T.H. Chan School of Public Health, Boston, MA, United States of America

* oluester2005@yahoo.com

## Abstract

In Nigeria, about 150000 babies are born annually with sickle cell disease (SCD), and this figure has been estimated to increase by 100% by the year 2050 without effective and sustainable control strategies. Despite the high prevalence, newborn screening for SCD which allows for early prophylactic treatment, education of parents/guardians and comprehensive management is not yet available. This study explored a strategy for screening in early infancy during the first and second immunization visits, determined the prevalence, feasibility and acceptability of early infant screening for SCD and the evaluation of the HemoTypeSC diagnostic test as compared to the high-performance liquid chromatography (HPLC) gold standard. A cross-sectional study was conducted in two selected primary health care centres in Somolu local government area (LGA) in Lagos, Nigeria. Two hundred and ninety-one mother-infant pairs who presented for the first or second immunization visit were consecutively enrolled in the study following written informed consent. The haemoglobin genotype of mother-infant pairs was determined using the HemoTypeSC rapid test kit. Confirmation of the infants' Hb genotype was done with HPLC. Data were analysed with SPSS version 22. Validity and Predictive value of HemotypeSC rapid screening test were also calculated. Infant screening for SCD was acceptable to 86% of mothers presenting to the immunization clinics. The prevalence of SCD among the infant cohort was 0.8%. The infants diagnosed with SCD were immediately enrolled in the paediatric SCD clinic for disease-specific care. The HemoTypeSC test had 100% sensitivity and specificity for sickle cell disease in early infancy compared to HPLC. This study affirms that it is feasible and acceptable for mothers to implement a SCD screening intervention program in early infancy in Lagos State. The study also demonstrates the utility of the HemotypeSC rapid testing for ease and reduced cost of screening infants for SCD.

**Data Availability Statement:** All relevant data are within the manuscript and its Supporting Information files.

**Funding:** EO This research was supported by the Fogarty International Center of the National Institutes of Health under Award Number D43TW010134. The content is solely the responsibility of the authors and does not necessarily represent the official views of the National Institutes of Health. The funders had no role in study design, data collection and analysis, decision to publish, or preparation of the manuscript.

**Competing interests:** The authors have declared that no competing interests exist.

## Introduction

Sickle cell disease (SCD) is an autosomal recessive genetically transmitted hemoglobinopathy responsible for significant morbidity and mortality [1, 2]. The disease affects red blood cells and is characterized by chronic haemolytic anaemia with several clinical consequences [3, 4]. It has been documented that 50–90% of children born with SCD do not reach their fifth birthday due to lack of diagnosis and comprehensive care [5, 6]. The burden of SCD in sub-Saharan Africa is expected to increase to 88% of worldwide cases by 2050 but despite the high prevalence, newborn screening programmes are not widely available [7–9].

In 2006, the World Health Organization (WHO) acknowledged SCD as a disease with high global impact and a remarkable public health significance in Africa with a greater need for attention to improve the overall child survival rate [10]. About 465,000 babies are born yearly with significant disorders of haemoglobin (Hb) of which 401,000 are SCD worldwide [7].

In Nigeria, the prevalence of sickle cell trait is about 23.7%, while the frequency of sickle cell disease is about 20 per 1000 births resulting in about 150000 babies being born annually with SCD. This figure ranks Nigeria, a country with the largest burden of SCD globally with about 2.69–5% of the population being affected [11, 12]. Also, the figure has been proposed to increase by 100% by the year 2050 without effective and sustainable control strategies [5, 7].

Currently, there is no national neonatal screening policy for diagnosis and management of SCD in Nigeria, most people with SCD are identified when they present with symptoms and the diagnosis confirmed by qualitative electrophoresis, which is most often available at the teaching hospitals and usually very expensive [13]. The devastating complications of SCD and lack of comprehensive health care contribute significantly to morbidity and mortality of persons with SCD. However, in high-income countries where the infrastructure for universal newborn screening, early intervention and comprehensive care exist, mortality and morbidity of SCD have reduced during the first 20 years of life, with less than 1% global disease burden and more than 90% of babies born with SCD survive into adulthood [5, 14, 15].

Early screening of infants for SCD allows for early initiation of prophylactic treatment, education of parents/guardians and comprehensive management leading to a reduction in mortality [11]. Evidence of multiple benefits of universal newborn screening for SCD in high prevalence regions has been demonstrated [9, 16]. In Jamaica, neonatal screening for SCD has led to improved outcomes in the affected babies, while in Brazil, newborn screening for SCD has resulted in about 300,000 babies been screened across 36 municipalities by the end of the year 2000 [17, 18].

Coordinated newborn screening for SCD has not been implemented in Nigeria [11]. Some of the challenges for initiation of the newborn programme in the country include; the current method of diagnosis which is laboratory-based Hb electrophoresis, iso-electric focusing, high-performance liquid chromatography (HPLC), mass spectrometry and molecular techniques. All of which involves lots of money, specially trained personnel and constant power source, which are not readily available. Besides, another important challenge of SCD diagnosis in Nigeria is the use of alkaline hemoglobin electrophoresis which is the most commonly available method of diagnosis, with lots of misdiagnosis of hemoglobin (Hb) genotypes [19, 20]. Hence, the need for an inexpensive, reliable, and easy to use point of care (POCT) devices with high sensitivity and specificity in the detection of Hb genotypes.

HemoTypeSC™ (Silver Lake Research Corp., USA) is a point of care test (POCT) kit intended for in vitro diagnostic use by health professionals to determine the presence of haemoglobin A, S and C in whole blood. It is based on competitive lateral flow immunoassay incorporating monoclonal antibodies for determination of the presence of haemoglobins A, S, and C to provide rapid detection of haemoglobin genotypes HbAA, HbSS, HbSC, HbCC, HbAS, and HbAC. HemoTypeSC has many advantages which include the ease of use, rapid results delivery and early

notification of the patients with necessary counselling. It can also be used in remote sites with early diagnosis resulting in reductions in mortality and morbidity due to SCD [19].

A multi-centre study in Nigeria which assessed a low-cost POCT device, HemoTypeSC reported the sensitivity and specificity of 93.4% and 99.9% respectively for the test [19]. Another study that tested the feasibility of implementing a sickle cell disease screening programme using innovative point-of-care test devices into existing immunisation programmes in primary health-care settings in Nigeria reported 100% sensitivity and specificity of HemotypeSC test in the detection of SCD [21]. Similarly, a study on point-of-care screening for SCD in low-resource settings reported 100% sensitivity and specificity for SCD [22].

The importance of carrying out pilot studies on the initiation of strategic advocacy initiatives to educate people about the benefits of newborn screening (NBS) for the successful implementation of NBS programme has been recommended [23, 24]. Moreover, despite the high prevalence of SCD in Nigeria, only 36% of all births take place in health facilities [24], resulting in difficulty in newborn screening. We developed strategies for screening early during the first or second immunization visits when the infants are brought to government primary health centres (PHCs) located in the communities. This study piloted an infant screening program for SCD during immunization visits in a Local Government Area (LGA) of Lagos State, Nigeria to determine the feasibility and acceptability of early infant screening. We also determined the prevalence of SCD among infants studied and evaluation of the HemoTypeSC diagnostic test as compared to a gold standard.

## Materials and methods

### Study location

Nigeria accounts for about half of West Africa's population with approximately 202 million people. It is a multi-ethnic and culturally diverse federation which consists of 36 autonomous states and the federal capital territory (Abuja) [25]. The literacy rate is 59.3 per cent for women and 79.9 per cent for men age 15–24 years. The rate is, however, low in the Northern region of Nigeria [26]. A report about poverty and inequality by the National Bureau of Statistics (NBS) from 2018 to October 2019, stated that 40% of people in Nigeria live below its poverty line of 137, 430 Naira ($381.75) a year, this represents about 82.9 million people.

The Nigerian healthcare system is organized into primary, secondary and tertiary healthcare levels. The local government areas (LGAs) are responsible for primary healthcare, the state governments are responsible for providing secondary care while the federal government is responsible for policy development, regulation, overall stewardship and provision of tertiary care. Household out-of-pocket expenditure (OOP) has remained the major source, constituting about 70.3% of total healthcare expenditure (THE) in 2009. Government expenditure on health as a percentage of GDP is below the average for sub-Saharan Africa. Less than 5% of Nigerians were covered by any form of health insurance at the end of 2013 [27].

Lagos State is in the Southwest geopolitical zone of Nigeria and the economic capital of Nigeria. This study took place in one (Somolu LGA) of the twenty local government areas (LGAs) [28] in Lagos state. Somolu LGA is a cosmopolitan community with a mixture of Nigerian ethnic groups dominated by the Yoruba ethnic group. It has a projected population of 1,361,100 as of 2015, an area of 10.3 km$^2$ and density of 132,190/km$^2$ [29].

### Study population, design, sample size determination and selection of participants

The study population included mothers and their infants aged between 2 to 10 weeks of age attending routine immunization clinics in two selected of the ten primary health centres (PHC) in the LGA. The study was cross-sectional and descriptive in design. Participants were

recruited among mothers who were 18 years and above and gave written informed consent. The Cochran formula for descriptive studies was used for sample size calculation ($n = z^2pq/d^2$) [30], with a standard normal deviation at 95% confidence interval (1.96), an estimated prevalence of SCD of 3% [12] and an error of precision at 2.5% (0.025). The minimum sample size calculated was 179 participants per group of mother-infant pairs. Considering a non-response rate of 20%, a sample size of 215 was calculated but 291 consenting pairs were eventually recruited for the study. All consenting mothers of infants presenting for first or second routine immunization visit postpartum between August 2019 and January 2020 were included while mothers of infants with a history of blood transfusion since birth were excluded.

## Ethical considerations

Ethical approval for this study was obtained from the Health Research and Ethics Committee (HREC) of the College of Medicine, University of Lagos *(CMUL/HREC/03/19/503)*. Written informed consent was obtained from each respondent with an assurance of confidentiality of the information and their right to withdraw from the study at any point in time. The participants were counselled to understand that study involvement was voluntary. The researchers ensured strict confidentiality of all participants' information. The blood samples were collected and sent for genotype confirmation at no cost to the participants and efforts were made to minimize discomfort to the participants during the blood sample collections.

## Pre-test counselling and recruitment of participants

Before the start of the study, training of research assistants, laboratory technicians and nurse counsellors was conducted. After an exhaustive review of the literature, we developed a research questionnaire to achieve the study objectives and pretested it among mother-infant pairs in a similar but different setting to the study location. Pre-test counselling was conducted for all consenting mothers by the trained nurse counsellors and informed consent for testing was obtained from the mother of each child before testing. The short questionnaire was then administered to obtain basic demographic information of mother-infant pairs and assess the acceptability of screening.

## Testing procedures

Five hundred microlitres of blood were collected from heel prick of each of the infants by trained technicians into EDTA microtainer tube from which 1–2 drops were collected for the HemoTypeSC while a fingerpick sample was collected from the mothers. HemoTypeSC™ rapid testing was performed immediately on-site after sample collection. The infants' blood samples were subsequently transported to the reference laboratory where the confirmatory HPLC method was employed. HemoTypeSC was compared with Bio-Rad D-10™ high-performance liquid chromatography haemoglobin testing system as a reference method to determine the accuracy of HemoTypeSC in detecting Hb genotypes AA (normal), AS (HbS trait), AC (HbC trait), SS (sickle cell anemia), SC (sickle-HbC disease), and CC (Hb C disease) at the Hemoglobin Reference Laboratory in the Sickle Cell Disease Foundation Nigeria (SCFN) Lagos, Nigeria.

## Post-test counselling and feedback to participants

The results of HemoTypeSC testing were communicated to all mothers after a post-test counselling session on the same clinic day while the results of the confirmatory test were communicated by phone within one week where there was no discordant result. Mothers and infants

confirmed to have SCD were invited back to the immunization clinic for further counselling and immediately referred for enrollment into the paediatrics sickle cell disease programme at Lagos University Teaching Hospital (LUTH).

Feasibility of early infant screening for SCD was assessed by >80% acceptability rate among participants, ≥ 90% of participants' willingness to enrol the child in clinic immediately if confirmed to have SCD, >60% of participants' willingness to pay (US$1·50) for the rapid screening test, >50% of participants affordability of cost at (US$1·50) for the rapid screening test and at least 80% of SCD confirmed babies receiving the diagnosis results and initiating immediate specific care.

## Data analyses

Data were analyzed with Statistical Package for Social Sciences (SPSS) version 22. Descriptive analyses were performed; the proportion of infants with sickle cell anaemia (HbSS), heterozygous for HbS and HbC (HbSC), with sickle cell trait (HbAS), heterozygous for HbA and HbC (HbAC), and with normal haemoglobin (HbAA) were calculated. The sensitivity, specificity, positive and negative predictive value of HemoTypeSC compared with the "gold standard" HPLC were also calculated.

## Results

### Socio-demographic characteristics of mother-infant pairs

Two hundred and ninety-one (291) mother-infant pairs participated in the study. The mean ± SD age of mothers was 29.9±5.4 years. The majority (92%) of the mothers had a minimum of secondary level education and most (82.2%) were employed. Two hundred and forty-five (84.2%) of the infants were 5 weeks or older [Table 1].

### Feasibility and acceptability of early infant screening for SCD

Forty-one (14%) of 291 mothers refused to screen their infants giving an 86% acceptance rate for early infant screening for SCD in this study. Ninety-four (74.6%) of 126 mothers who refused to screen themselves knew their Hb genotype and 40(35%) stated their Hb genotype and that of their spouses as HbAA. Hence, they believed they could not have a child with SCD. The majority of mothers (92%) who accepted screening did so to know their genotype and that of their babies; 148(59%) did so also because the screening was free. Whereas only 23% of the mothers stated that soon after birth is the most appropriate time to screen, about 36% felt that the most appropriate time should be within the first month of birth while almost all, 283 (97.3%) stated their willingness to enrol their infants in SCD clinic immediately if found to have SCD after screening. Although the majority were willing 243(84%) and could afford 239 (82%), to pay ₦540.00 (US$1·50) for the rapid SCD screening test, only 190(65%) were willing to pay additional ₦7,000.00 (US$20) required for confirmatory testing and only 143(49%) could afford the cost. The perceived challenges for early infant screening by mothers in this study included the cost of confirmatory test 190(65%), availability of test facility 155(53%), possible delay or not getting the test result 149(51%) and accessibility for follow up care 148 (51%) among others [Table 2].

### Test results

Two hundred and fifty infants were screened for SCD with HemoTypeSC rapid test strips and Bio-Rad D-10™ high-performance liquid chromatography Hemoglobin testing system. Overall, the HemoTypeSC tests identified 185(74.0%) HbAA, 53(21.2%) HbAS, 9(3.6%) HbAC, 1

**Table 1. Socio-demographic characteristics of mother-infant pairs.**

| Socio-demographic characteristics | Frequency (N = 291) | Percentage (%) |
|---|---|---|
| **Age of mothers (Years)** | | |
| 18–28 | 123 | 42.3 |
| 29–38 | 145 | 49.8 |
| 39–48 | 23 | 7.9 |
| Mean ± SD = 29.9± 5.4years | | |
| **Marital status of mothers** | | |
| Single | 6 | 2.1 |
| Married | 285 | 97.9 |
| **Level of education of mothers** | | |
| None | 3 | 1.0 |
| Primary | 19 | 6.5 |
| Secondary | 137 | 47.1 |
| Tertiary | 132 | 45.4 |
| **Employment status of mothers** | | |
| Unemployed | 50 | 17.2 |
| Employed | 241 | 82.2 |
| **Registered for ANC* in pregnancy** | | |
| Yes | 287 | 98.6 |
| No | 4 | 1.4 |
| **Facilities attended for ANC*** | | |
| Tertiary hospital | 2 | 0.7 |
| General hospital | 52 | 17.9 |
| PHCs | 58 | 19.9 |
| Private hospital | 166 | 57.0 |
| Traditional Birth Attendants | 13 | 4.5 |
| **Age of infants (weeks)** | | |
| <5weeks | 46 | 15.8 |
| 5–10 weeks | 245 | 84.2 |
| Mean ± SD = 7.4 ± 2.3 weeks | | |
| **Gender of infants** | | |
| Male | 153 | 52.6 |
| Female | 138 | 47.4 |

*ANC- Antenatal clinic

(0.4%) HbSS, 1(0.4%) HbCC and 1(0.4%) HbSC. Thus, 2(0.8%) infants had sickle cell disease and 1(0.4%) had HbC disease. The sensitivity, specificity, positive and negative predictive values of HemoTypeSC compared to "gold standard" HPLC were independently calculated. The validity and predictive accuracy of HemoTypeSC screening test was calculated as follows [31];

Sensitivity refers to the ability of the rapid kit test to correctly identify positive SCD cases:

$$\text{sensitivity} = \frac{TP}{TP + FN} \text{ X } 100\%$$

Specificity refers to the ability of the rapid kit test to correctly identify negative SCD cases:

$$\text{Specificity} = \frac{TN}{TN + FP} \text{ X } 100\%$$

**Table 2. Feasibility and acceptability of early infant screening for SCD.**

| Feasibility and acceptability variables | Number of participants (N = 291) | Percentage (%) |
|---|---|---|
| **Mothers who accepted to be screened for Hb genotype** | 165 | 56.7 |
| **Children who were accepted to be screened for Hb genotype by mothers)** | 250 | 85.9 |
| **Reasons for wanting self/child to be screened**[*] | | |
| Just to know my genotype and that of my child | 231 | 91.7 |
| Have a child with SCD | 3 | 1.2 |
| Had a child with SCD but late | 0 | 0.0 |
| My spouse and I are both carriers of SCD haemoglobin | 3 | 1.2 |
| The screening is free for me and my child | 148 | 59.0 |
| Other reason | 7 | 3.8 |
| **Reasons for not wanting self/child to be screened**[*] | | |
| I know my genotype | 94 | 80 |
| I know my child's genotype | 3 | 2.6 |
| My husband and I are both AA | 40 | 34.5 |
| To avoid being worried | 4 | 3.4 |
| Don't want my child to be pricked/bled | 27 | 23.3 |
| Will take time for the result to be ready | 7 | 6.1 |
| Do not like free test | 3 | 2.7 |
| Other reasons | 8 | 9.1 |
| **Appropriate time a child should be screened for Hb** | | |
| During pregnancy | 38 | 13.0 |
| Soon after birth | 67 | 23.0 |
| Within the first one-month of birth | 105 | 36.1 |
| Pre-school | 43 | 14.8 |
| Others | 20 | 6.9 |
| Don't know | 18 | 6.2 |
| Willingness to enrol the child in SCD clinic immediately if SS/SC after testing | 283 | 97.3 |
| Willingness to pay ₦540.00(US$1·50) for rapid screening test | 243 | 83.5 |
| Affordability to pay ₦540.00(US$1·50) for rapid screening test | 239 | 82.1 |
| Willingness to pay ₦7,000.00 (US$20) for confirmatory Hb test) | 190 | 65.3 |
| Affordability to pay ₦7,00.00 (US$20) for confirmatory Hb test | 143 | 49.1 |
| **Perceived challenges for early infant screening for Hb**[*] | | |
| Cost of the confirmatory test at ₦7,000.00 (US$20) | 190 | 65.3 |
| Availability of test facility | 155 | 53.3 |
| Delay in getting the test result | 149 | 51.2 |
| Availability/accessibility for follow up care | 148 | 50.9 |
| Accessibility to test result | 147 | 50.5 |
| Time committed to counselling and testing | 147 | 50.5 |
| Cost of screening test at ₦540.00(US$1.50) | 101 | 35.3 |
| Fear of knowing the SCD status of the child | 86 | 29.6 |
| Other reasons | 22 | 9.6 |

[*] multiple answers applied

Positive Predictive Value (PPV) refers to the probability that a participant with a positive result truly has SCD:

$$PPV = \frac{TP}{TP + FP} X\ 100\%$$

Negative Predictive Value (NPV) refers to the probability that a participant with a negative result truly does not have SCD:

$$NPV = \frac{TN}{TN + FN} X\ 100\%$$

HemoTypeSC showed entirely consistent results with HPLC with a sensitivity and specificity of 100%. HemoTypeSC correctly identified every HbSS and HbCC genotype with no false-positive results, exhibiting sensitivity and specificity of100% [Table 3]. The mothers of the two confirmed infants who were diagnosed with SCD were known carriers of an abnormal haemoglobin variant (HbAS, HbAC). Both mothers and the mother of the child with Hb CC received the diagnosis results, were counselled and referred to paediatric sickle cell disease clinic for specific care [Table 3].

## Discussion

This study demonstrated that early infant (≤10 weeks) screening for SCD using point of care testing (POCT) at the primary health care centres is feasible. It has the potential to be implemented on a large scale and could be fully integrated into the National Programme on immunization (NPI) with little additional resources.

In developed countries, universal newborn/early infant diagnosis of SCD has been estimated to reduce mortality in 94–99% of children with SCD with most now living to adulthood [15, 32, 33]. Newborn screening leads to a reduction in the morbidity and mortality through early identification and commencement of definitive care including early initiation of prophylactic penicillin, pneumococcal vaccination, counselling for the parents, and other health management [34]. In Nigeria, due to the lack of a coordinated national universal newborn screening program, children with SCD are mostly diagnosed at the onset of sickle cell complication(s). Although the federal government of Nigeria initiated a newborn screening (NBS) program with the creation of six comprehensive treatment centres in each of the six geopolitical zones, the effort to establish universal newborn screening has not been successful despite the significant investment by the government [23, 35].

**Table 3. Validity and predictive accuracy of HemoTypeSC screening test.**

| Hb | HemoTypeSC | HPLC | HemoTypeSC | HemoTypeSC | HemoTypeSC | HemoTypeSC |
|---|---|---|---|---|---|---|
| | Freq. (%) | Freq. (%) | Sensitivity | Specificity | PPV | NPV |
| | | | TP/(TP+FN) * | TN/(FP+TN) * | TP/(TP+FP) * | TN/(TN+FN) * |
| AA | 185 (74.0%) | 185 (74.0%) | 185/185 = 100% | 65/65 = 100% | 185/185 = 100% | 65/65 = 100% |
| AS | 53 (21.2%) | 53 (21.2%) | 53/53 = 100% | 197/197 = 100% | 52/52 = 100% | 197/197 = 100% |
| AC | 9(3.6%) | 9 (3.6%) | 9/9 = 100% | 241/241 = 100% | 9/9 = 100% | 241/241 = 100% |
| SS | 1 (0.4%) | 1 (0.4%) | 1/1 = 100% | 249/249 = 100% | 1/1 = 100% | 249/249 = 100% |
| SC | 1 (0.4%) | 1 (0.4%) | 1/1 = 100% | 249/249 = 100% | 1/1 = 100% | 249/249 = 100% |
| CC | 1 (0.4%) | 1 (0.4%) | 1/1 = 100% | 249/249 = 100% | 1/1 = 100% | 249/249 = 100% |

*TP (true positive); FN (false negative); TN (true negative); FP (false positive) PPV (positive predictive value); (NPV (negative predictive value)

The prevalence of SCD among the infant cohorts in this study was 0.8%, while that of carrier traits HbAS and HbAC were 21.2% and 3.6% respectively. One hundred and eighty-five (74%) of the infants had their haemoglobin genotypes as AA. A similar study in Abuja, Nigeria reported a prevalence of 1·4% for HbSS, 20·5% for HbAS and HbAA 77% [21]. Another similar study in Awka South-East Nigeria reported a prevalence of 0.3% for HbSS and 75% for HbAA genotype amongst newborns [36]. A study in Benin, Southern Nigeria found 75.3% HbAA, 20.6% HbAS, 1.1% HbAC, 2.8% HbSS, and 0.2% HbSC [12]. While another study in Northern Nigeria reported a prevalence of 2.69% for SCD [5]. The finding of our study corroborates that of the DHS report of 2018 which reported 20% HbAS, 1% HbSS and HbSC. Variation in the prevalence of SCD according to the geographic region has been reported by the demographic and health survey (DHS) 2008 is highest in the Southwest (2%) and lowest in the South-South (0.3%) [37].

Our study showed high acceptability of early infant screening 250(86%) among the respondents. This finding is similar to that of a multi-centre survey of acceptability of newborn screening for SCD in Nigeria which reported 86.1% [35]. Similarly, studies in South-western and Northern Nigeria confirmed the acceptability of newborn/infants SCD diagnosis [5, 12]. Higher rates (99%) have been reported by a study in Liberia [38], and a Ghanaian study has documented that screening and follow-up of newborns for SCD is feasible in developing countries in Africa [39]. These findings indicate that implementation of universal early infant screening for SCD is highly likely to be acceptable within the country.

All the infants diagnosed with SCD were immediately enrolled in the SCD clinic for disease-specific care in this study. The mothers of the two infants who were diagnosed with SCD in this study were known carriers of an abnormal haemoglobin variant (HbAS, HbAC) but stated their spouses Hb status as HbAA. Hence, they did not expect to have had a child with an abnormal haemoglobin variant. Thus, universal early infant screening which allows for diagnosis and initiation of preventative therapies before the onset of symptoms is of utmost importance.

This study demonstrates the feasibility of incorporating early infants screening programme into the routine immunization programmes in primary health-care centres which will promote accessibility and sustainability. Immunization as a component of primary health care is sustainable and this can be taken advantage of for the establishment of an early infant screening programme for SCD. The approach used in this study with PHCs appears to have encouraged the feasibility of the screening programme.

The majority of the mothers access immunization services for their infants at the primary rather than the secondary and tertiary centres which have longer waiting times and higher costs [35]. The Hb genotype screening test was provided along with the routine immunization and not as a standalone service. The staff members of the PHCs were involved in the programme which also reduced the cost of services. The majority of the mothers in this study showed a willingness to pay for the point of care test which was relatively cheap (US$1.5) compared to the standard test method (US$20). The low-cost rapid point-of-care test and other factors suggest the feasibility of early infants screening for SCD in Nigeria and other developing countries. Also, the low cost of screening most likely will encourage coverage by insurance schemes which will further enhance accessibility and sustainability. For a feasible and sustainable early diagnosis of SCD and enrollment into care programmes policy, the use of simple, affordable and appropriate technology which can be applied at the PHCs level needs to be considered for the benefits of large proportions of the community [19].

HemoTypeSC exhibited 100% sensitivity and specificity for detection of each of three Hb phenotypes, HbA, HbS, and HbC in our study. This finding supports other documented laboratory reports, which reported a sensitivity and specificity of 100% for HbS and HbC in ideal

conditions [19, 22]. HemoTypeSC is affordable, rapid, reliable and accurate diagnostic testing for early infants' screening programs for SCD in low-resource countries where the prevalence of SCD is high without an organized national newborn screening program [19]. A similar and larger study in Abuja Nigeria also reported a sensitivity and specificity of 100% with HemoTypeSC and have demonstrated point-of-care tests as reliable and accurate in newborn screening for SCD [21].

The cost of the confirmatory test, availability of test facility, delay in getting the test result, availability/accessibility for follow up care, accessibility to test result and time committed to counselling and testing were among the perceived challenges by mothers for early infant screening. Similar studies in various regions have documented high cost, logistical complexity of conventional diagnostic methods and the delayed availability of screening results as the limitation for the sustainability of newborn screening for SCD in sub-Saharan Africa and other resource-limited regions worldwide [16, 40, 41]. A study in Nigeria found the main barriers to SCD newborn screening were likely to be financial and practical, rather than social or cultural factors [35].

The challenges stated by the participants in our study were mitigated by high sensitivity and specificity and low cost of the POCT, rapid result and the ready access to the primary health care facility. Hence, we recommend setting up routine regular basic SCD care at the PHC, an annual evaluation at a secondary care facilities level of care and only complex and complicated care at tertiary health care centres.

## Conclusion

This study results showed the feasibility and acceptability of early infant screening for SCD using simple, affordable and appropriate technology that can be adapted at a community level to ensure early diagnosis and prompt referral for management to reduce the burden of SCD in Nigeria. This study also demonstrated that HemoTypeSC is an affordable, rapid, and accurate diagnostic testing for early infant screening programs for SCD at PHC in developing countries. Therefore, governments need to provide low cost, equitable and promote access for early screening and diagnosis of SCD within the community.

## Supporting information

**S1 Appendix. Study questionnaire.**
(PDF)

**S2 Appendix. Study data set.**
(XLS)

**S3 Appendix. IRB approval.**
(PDF)

## Acknowledgments

The authors acknowledge the participants, medical officer of health (MOH) of the LGA, Ajoke Oyetunji of SCFN and the entire research team.

## Author Contributions

**Conceptualization:** Esther O. Oluwole, Titilope A. Adeyemo, Gbemisola E. Osanyin, Oluwakemi O. Odukoya, Phyllis J. Kanki, Bosede B. Afolabi.

**Data curation:** Esther O. Oluwole, Titilope A. Adeyemo.

**Formal analysis:** Esther O. Oluwole.

**Funding acquisition:** Esther O. Oluwole.

**Investigation:** Esther O. Oluwole, Bosede B. Afolabi.

**Methodology:** Esther O. Oluwole, Oluwakemi O. Odukoya, Bosede B. Afolabi.

**Project administration:** Esther O. Oluwole, Titilope A. Adeyemo, Gbemisola E. Osanyin, Oluwakemi O. Odukoya, Bosede B. Afolabi.

**Resources:** Esther O. Oluwole.

**Supervision:** Esther O. Oluwole, Titilope A. Adeyemo, Gbemisola E. Osanyin, Bosede B. Afolabi.

**Validation:** Esther O. Oluwole, Titilope A. Adeyemo, Bosede B. Afolabi.

**Visualization:** Esther O. Oluwole.

**Writing – original draft:** Esther O. Oluwole, Titilope A. Adeyemo.

**Writing – review & editing:** Esther O. Oluwole, Titilope A. Adeyemo, Gbemisola E. Osanyin, Oluwakemi O. Odukoya, Phyllis J. Kanki, Bosede B. Afolabi.

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
