## [Decision Letter · Decision Letter 0]

23 Sep 2020

PONE-D-20-26306

Feasibility and acceptability of early infant screening for sickle cell disease in Lagos, Nigeria------A pilot study

PLOS ONE

Dear Dr. OLUWOLE

Thank you for submitting your manuscript to PLOS ONE. After careful consideration, we feel that it has merit but does not fully meet PLOS ONE’s publication criteria as it currently stands. Therefore, we invite you to submit a revised version of the manuscript that addresses the points raised during the review process.

We look forward to receiving your revised manuscript.

Kind regards,

Ambroise Wonkam, MD, PhD

Academic Editor

PLOS ONE

Journal Requirements:

3. Please include additional information regarding the survey or questionnaire used in the study and ensure that you have provided sufficient details that others could replicate the analyses.

For instance, if you developed a questionnaire as part of this study and it is not under a copyright more restrictive than CC-BY, please include a copy, in both the original language and English, as Supporting Information.

4. In the Methods, please discuss whether and how the questionnaire was validated and/or pre-tested. If this did not occur, please provide the rationale for not doing so.

Additional Editor Comments:

Dear Authors,

Your paper addresses and important question of public health importance in Africa, and globally: the screening of Sickle cell disease.

Although the authors have a genuine attempt to address your request question, in its present form, the methodology and quality of the reporting need some major improvement as indicated by both reviewers.

Moreover, more background information on the setting (Nigeria) are needed to put the paper into context.

Population size? SCD incidence and prevalence? General level of formal education? Poverty indices? National health system? Available public health policies and current practices regarding SCD screening; Availability and access to comprehensive services for SCD prevention and care.

Please carefully address point-to-point the editor and reviewers’ comments, as this is expected to improve the quality of your text.

Reviewers' comments:

Reviewer's Responses to Questions

**Comments to the Author**

1. Is the manuscript technically sound, and do the data support the conclusions?

Reviewer #1: Yes

Reviewer #2: Yes

2. Has the statistical analysis been performed appropriately and rigorously? 

Reviewer #1: Yes

Reviewer #2: No

3. Have the authors made all data underlying the findings in their manuscript fully available?

Reviewer #1: Yes

Reviewer #2: Yes

4. Is the manuscript presented in an intelligible fashion and written in standard English?

Reviewer #1: No

Reviewer #2: Yes

5. Review Comments to the Author

Reviewer #1: This is an interesting and useful article describing the authors’ experience studying the feasibility and acceptability of early infant screening for sickle cell disease among mothers and their newborns attending a community health center in Lagos for routine immunization. It also reports on the ease of use and accuracy of HemoTypeSC compared to HPLC in determining Hb genotypes among young infants. They studied 291 mother/infant pairs and ~86% of the mothers accepted screening. The breakdown of Hb genotypes was in agreement with previous reports from Nigeria. Moreover, HemoTypeSC was easy to use and showed 100% specificity and sensitivity in identifying those with abnormal Hb genotypes. Thus, this article builds on previous positive reports for HemoTypeSC in other parts of Nigeria.

While the study was well designed and executed, the style of writing needs considerable improvement.

In general terms, I would advise the authors to refer to the diagnoses as Hb genotypes instead of phenotypes so as not to be confused with clinical phenotypes.

There’s need to include in the introduction, the justification for using POCT in newborn screening in preference to other available screening methods especially in resource-poor areas. A brief mention of previous work with HemoTypeSC in Nigeria and other places is also apropos.

There are many grammatical and syntax issues and the authors are advised to review the whole manuscript carefully in this regard. Some suggested corrections are outlined below:

Abstract:

Line 31, should read “……comprehensive management is not…..”

Line 39, should read “The haemoglobin genotype of mother-infant pairs…..”

Line 41, should read “Data were analysed with SPSS version 22”.

Introduction: The whole of the introduction needs to be restructured. Currently it just one long paragraph and it does not flow well. Apart from briefly introducing sickle cell disease epidemiology, it should focus on the problems associated with newborn screening and the current methods as mentioned above

Line 59 - 62, “In high income countries where the infrastructure for universal newborn screening, early intervention and comprehensive care exist, mortality has reduced from 16% to <1% with less than 1% global disease burden and over 90% of babies born with SCD surviving into adulthood”. It is not clear if the 16 to 1% figure is referring to overall mortality or only in childhood.

Materials and Methods

Lines 88 – 90. The sentence “The State has the highest population in Nigeria with twenty local government areas (LGA), and this study took place in one of them which was selected by simple random sampling through ballot” needs to be restructured.

Lines 127 – 133. Most of the description of HemoTypeSC given here rightly belongs in the introduction.

Line144, should read: “…heelprick of each of the infants by trained ….”

Line 147, The infants’ blood samples were….”

Line 153, should read “…. while the results of the confirmatory test were communicated….”

Line157 – 162. This section, starting with “Feasibility..” should be in a new paragraph. Moreover, it is not clear how the authors arrived at the cut-off figures quoted. Are they from previous studies? If so, the source should be referenced; otherwise their basis should be explained.

Results:

Line 174, should read, “Socio-demographic characteristics of mother-infant pairs”.

Line 206, should read, “Ninety-four (74.6%) of 126 mothers who refused….”

Line 218, delete “the cost of”

Table 2: The title of the second column is not clear. It is certainly not frequency, but number (n) of individuals affected. The total of 291 can be moved to the title of the table. The section on “Reasons for not wanting self/child to be screened” should be in a separate table since the denominator is not 291, but the 126 that did not agree to be screened. It this is true, the percentages given in the table cannot be correct.

Discussion:

Apart from the accuracy of HemoTypeSC and its low cost, the other disadvantages of HPLC in a resource-poor environment should be stressed. These include the need for highly skilled personnel, reagents and electrical power, which, is quite often, not available.

Line 292 - 293, should read “….with the creation of a comprehensive treatment center in each of the 6 geopolitical zones….”

Line 305, what is DHS?

Line 316, should read “The mothers of the two infants who were diagnosed…..”

Line 330, a new paragraph should begin with the sentence, “The majority of the mothers…..”

Reviewer #2: Review of Feasibility and acceptability of early infant screening for sickle cell disease in Lagos, Nigeria

This paper appears to have two objectives, an evaluation of the HemoType SC diagnostic test and an assessment of haemoglobinopathy detection in primary health care centres. The sample population was drawn from 2 of 10 health centres in the Somolu Community of Lagos. Reference is made to a paper reporting that only 36% deliveries take place in health facilities in Nigeria yet 95% of the studied population delivered in hospitals/health centres. It would also be helpful to know what proportion of births attend ‘routine immunization clinics’. Although 291 mother/infant ‘units’ were admitted to the study, 41 (14%) refused screening and the principle reason appears to be that they believed that they and their spouses had an AA genotype. Using the Bio-Rad D-10 as a reference method, the performance of the diagnostic kit appears acceptable but the numbers are far too small for reliable assessment with the detection of only one SS, one SC and I CC baby. Overall 53 mothers were believed to have the sickle cell trait but the fathers of 2 infants with sickle cell disease were believed by the mother to have an AA phenotype. This new information should be presented in the results and not in the discussion, along with the possible explanation – were these beta thalassaemia traits, not the fathers, or assumed to be normal because they were well. The difference between ‘willingness’ and ‘affordability’ in Table 2 was not clear to this reviewer. Overall the paper is far too long and poorly organized, the numbers too small for reliable conclusions and with 95% deliveries occurring in hospital/health care centres, it seems that greater effort should be put into sample collection at birth.

6. PLOS authors have the option to publish the peer review history of their article (what does this mean?). If published, this will include your full peer review and any attached files.

Reviewer #1: No

Reviewer #2: **Yes: **Graham R Serjeant

---

## [Author Response · Author response to Decision Letter 0]

25 Oct 2020

Editor Comments:

Your paper addresses and important question of public health importance in Africa, and globally: the screening of Sickle cell disease.

Although the authors have a genuine attempt to address your request question, in its present form, the methodology and quality of the reporting need some major improvement as indicated by both reviewers.

Moreover, more background information on the setting (Nigeria) are needed to put the paper into context.

Population size? SCD incidence and prevalence? General level of formal education? Poverty indices? National health system? Available public health policies and current practices regarding SCD screening; Availability and access to comprehensive services for SCD prevention and care.

Please carefully address point-to-point the editor and reviewers’ comments, as this is expected to improve the quality of your text.

Author’s Comments:

Thank you, sir, for your comments.

The methodology and quality of the reporting has been reviewed with major improvement as indicated by the reviewers. Also, more background information on the setting (Nigeria) have been added to put the paper into context in the method section. (Page 6-7; Lines 134-148)

We have carefully addressed the point-to-point the editor and reviewers’ comments, to improve the quality of the manuscript.

Reviewer #1 Original comments of the reviewer Reply by the author(s)/ Changes Made Changes done on page number and line number

General In general terms, I would advise the authors to refer to the diagnoses as Hb genotypes instead of phenotypes so as not to be confused with clinical phenotypes. Thank you, Sir, for all the comments. 

This has been corrected throughout the manuscript All pages

Introduction There’s need to include in the introduction, the justification for using POCT in newborn screening in preference to other available screening methods especially in resource-poor areas. A brief mention of previous work with HemoTypeSC in Nigeria and other places is also apropos.

 These has been included in the introduction. Pages 4-5

Lines 90-99

 There are many grammatical and syntax issues and the authors are advised to review the whole manuscript carefully in this regard. Some suggested corrections are outlined below: Grammatical and syntax issues have been addressed carefully. All pages

Abstract:

 Line 31, should read “……comprehensive management is not…..”

 Corrected Pg. 2; Line 33

 Line 39, should read “The haemoglobin genotype of mother-infant pairs…..” Corrected Pg. 2; Line 41

 Line 41, should read “Data were analysed with SPSS version 22”. Corrected Pg. 2; Line 43

 The whole of the introduction needs to be restructured. Currently it just one long paragraph and it does not flow well. Apart from briefly introducing sickle cell disease epidemiology, it should focus on the problems associated with newborn screening and the current methods as mentioned above

Line 59 - 62, 

“In high income countries where the infrastructure for universal newborn screening, early intervention and comprehensive care exist mortality has reduced from 16% to <1% with less than 1% global disease burden and over 90% of babies born with SCD surviving into adulthood”. It is not clear if the 16 to 1% figure is referring to overall mortality or only in childhood.

DONE accordingly

This statement has been rephrased Pages 3-6;

 Lines 53-128

Pg.4; 

Lines 77-80

Materials and Methods 

 Lines 88 – 90. The sentence “The State has the highest population in Nigeria with twenty local government areas (LGA), and this study took place in one of them which was selected by simple random sampling through ballot” needs to be restructured. 

The statement has been restructured Pg. 7

Line 150-151

 Lines 127 – 133. Most of the description of HemoTypeSC given here rightly belongs in the introduction.

 The description of HemoTypeSC has been moved to the introduction.

 Pg. 5

Line 101-108

 Line144, should read: “…heelprick of each of the infants by trained ….”

 Corrected Pg. 8;

Line 188

 Line 147, The infants’ blood samples were….” Corrected Pg. 9;

Line 191

 Line 153, should read “…. while the results of the confirmatory test were communicated….” Corrected Pg. 9;

Line 201

 Line157 – 162. This section, starting with “Feasibility.” should be in a new paragraph. 

Moreover, it is not clear how the authors arrived at the cut-off figures quoted. Are they from previous studies? If so, the source should be referenced; otherwise their basis should be explained. This section has been moved to a new paragraph

The cut-off figures were the standard set for the feasibility before the study was conducted. These data shows that the intervention is practical and feasible.

They were not quoted from previous study Pg. 9;

Lines 207-212

Results:

 Line 174, should read, “Socio-demographic characteristics of mother-infant pairs”. Corrected Pg. 10;

Line 231

 Line 206, should read, “Ninety-four (74.6%) of 126 mothers who refused….” Corrected Pg. 12;

Line 270

 Line 218, delete “the cost of” Deleted 

 Table 2: The title of the second column is not clear. It is certainly not frequency, but number (n) of individuals affected. The total of 291 can be moved to the title of the table. Frequency has been changed to number of participants (n) 

The total number of participants in each column/row has been indicated 

Pg. 13;

Line 291

 The section on “Reasons for not wanting self/child to be screened” should be in a separate table since the denominator is not 291, but the 126 that did not agree to be screened. It this is true, the percentages given in the table cannot be correct. Yes, 126 participants did not want their child to be screened. But the question referred to reasons why both mothers and child refused screening. Not only the infants. Moreover, the question had multiple responses and participants had opportunity to pick more than one answer. This was indicated by the sign (*) with the explanation given below the table.

 Table 2; page 13

Discussion:

 Apart from the accuracy of HemoTypeSC and its low cost, the other disadvantages of HPLC in a resource-poor environment should be stressed. These include the need for highly skilled personnel, reagents and electrical power, which, is quite often, not available. This point has been stated in the introduction. To avoid repetition in the discussion. Pg. 4;

Lines 90-99

 Line 292 - 293, should read “…with the creation of a comprehensive treatment center in each of the 6 geopolitical zones….” Corrected Pg. 16; 

Line 355

 Line 305, what is DHS? demographic and health survey. Now included Pg. 17; 

Line 368

 Line 316, should read “The mothers of the two infants who were diagnosed…..” Corrected Pg. 17; 

Line 379

 Line 330, a new paragraph should begin with the sentence, “The majority of the mothers…..” Done. Pg. 18; 

Line 391

Reviewer #2: 

 Reference is made to a paper reporting that only 36% deliveries take place in health facilities in Nigeria yet 95% of the studied population delivered in hospitals/health centres. Thank you, sir, for this point.

The 95% in this study only attended antenatal clinic in hospital/health facilities but not delivered in same.

 Table 1; pg.11

---

## [Editor Report · Decision Letter 1]

11 Nov 2020

Feasibility and acceptability of early infant screening for sickle cell disease in Lagos, Nigeria------A pilot study

PONE-D-20-26306R1

Dear Dr.  OLUWOLE

We’re pleased to inform you that your manuscript has been judged scientifically suitable for publication and will be formally accepted for publication once it meets all outstanding technical requirements.

Kind regards,

Ambroise Wonkam

Academic Editor

PLOS ONE
---

## [Editor Report · Acceptance letter]

23 Nov 2020

PONE-D-20-26306R1 

Feasibility and acceptability of early infant screening for sickle cell disease in Lagos, Nigeria------A pilot study 

Dear Dr. OLUWOLE:

I'm pleased to inform you that your manuscript has been deemed suitable for publication in PLOS ONE. Congratulations! Your manuscript is now with our production department. 

Kind regards, 

on behalf of

Professor Ambroise Wonkam 

Academic Editor

PLOS ONE